# Maternal and Neonatal Characteristics and Outcomes of COVID-19 in Pregnancy: An Overview of Systematic Reviews

**DOI:** 10.3390/ijerph18020596

**Published:** 2021-01-12

**Authors:** Michail Papapanou, Maria Papaioannou, Aikaterini Petta, Eleni Routsi, Maria Farmaki, Nikolaos Vlahos, Charalampos Siristatidis

**Affiliations:** 1Second Department of Obstetrics and Gynecology, “Aretaieion Hospital”, Medical School, National and Kapodistrian University of Athens, Vas. Sofias 76, 11528 Athens, Greece; mixalhspap13@gmail.com (M.P.); mariapp1nn@gmail.com (M.P.); cathpetta@hotmail.com (A.P.); routsie@gmail.com (E.R.); mfarmaki19@gmail.com (M.F.); nikosvlahos@med.uoa.gr (N.V.); 2Assisted Reproduction Unit, Second Department of Obstetrics and Gynecology, “Aretaieion Hospital”, Medical School, National and Kapodistrian University of Athens, Vas. Sofias 76, 11528 Athens, Greece

**Keywords:** COVID-19, SARS-CoV-2, pregnancy, mother, maternal outcomes, neonate, vertical transmission

## Abstract

(1) *Background*: A considerable number of systematic reviews, with substantial heterogeneity regarding their methods and included populations, on the impact of COVID-19 on infected pregnant women and their neonates, has emerged. The aim was to describe the obstetric-perinatal and neonatal outcome of infected pregnant women and their newborns during the COVID-19 pandemic; (2) *Methods*: Three bibliographical databases were searched (last search: 10 September 2020). Quality assessment was performed using the AMSTAR-2 tool. Primary outcomes included mode of delivery, preterm delivery/labor, premature rupture of membranes (PROM/pPROM) and abortions/miscarriages. Outcomes were mainly presented as ranges. A separate analysis, including only moderate and high-quality systematic reviews, was also conducted. The protocol was registered with PROSPERO (CRD42020214447); (3) *Results*: Thirty-nine reviews were analyzed. Reported rates, regarding both preterm and term gestations, varied between 52.3 and 95.8% for cesarean sections; 4.2–44.7% for vaginal deliveries; 14.3–63.8% specifically for preterm deliveries and 22.7–32.2% for preterm labor; 5.3–12.7% for PROM and 6.4–16.1% for pPROM. Maternal anxiety for potential fetal infection contributed to abortion decisions, while SARS-CoV-2-related miscarriages could not be excluded. Maternal ICU admission and mechanical ventilation rates were 3–28.5% and 1.4–12%, respectively. Maternal mortality rate was <2%, while stillbirth, neonatal ICU admission and mortality rates were <2.5%, 3.1–76.9% and <3%, respectively. Neonatal PCR positivity rates ranged between 1.6% and 10%. After accounting for quality of studies, ranges of our primary outcomes remained almost unchanged, while among our secondary outcomes, maternal ICU admission (3–10%) and mechanical ventilation rates (1.4–5.5%) were found to be relatively lower; (4) *Conclusions*: Increased rates of cesarean sections and preterm birth rates were found, with iatrogenic reasons potentially involved. In cases of symptomatic women with confirmed infection, high maternal and neonatal ICU admission rates should raise some concerns. The probability of vertical transmission cannot be excluded. Further original studies on women from all trimesters are warranted.

## 1. Introduction

The novel coronavirus disease 2019 (COVID-19) caused by the Severe Acute Respiratory Syndrome coronavirus 2 (SARS-CoV-2) has become a global health emergency since its declaration as a pandemic on 11 March 2020 by the World Health Organization [1]. SARS-CoV-2 can cause severe illness in older people with comorbidities, yet everyone can be infected, from adults, adolescents and children, up to pregnant women and neonates [2,3].

Experience from already known viral infections during pregnancy revealed increased maternal complications, such as spontaneous abortions, premature rupture of membranes and preterm labor [3]. Symptoms are similar to influenza and the rate of spread is greater than that for previous coronaviruses, such as the Severe Acute Respiratory Syndrome coronavirus (SARS-CoV) and the Middle East Respiratory Syndrome coronavirus (MERS-CoV), demonstrated that infected individuals could suffer from serious complications like acute respiratory distress syndrome and multi-organ failure [4]. It is well established that a viral pneumonia, especially when accompanied by comorbidities, including chronic cardiovascular and respiratory problems, and/or obesity can significantly increase maternal and neonatal morbidity [5].

Given the severe prognosis of pregnant women affected by SARS-CoV or MERS-CoV, many concerns have been raised about the effects of SARS-CoV-2 on such a sensitive group of patients [4]. This resulted in various studies being conducted across the world by different study groups, employing variable study populations, examining potential adverse maternal, fetal and neonatal outcomes. Based on the large number of systematic reviews that emerged since the beginning of the pandemic and the substantial heterogeneity of both methods and populations employed, and considering that these systematic reviews currently constitute the basis of clinical decisions in such a sensitive group of affected patients, we decided to carry out an overview of the existing literature. The aim of our study was to depict the obstetric-perinatal and neonatal outcome of infected pregnant women and their newborns during the pandemic, using the up-to-date published evidence.

## 2. Materials and Methods

### 2.1. Eligibility Criteria

Inclusion criteria were as follows: (1) systematic reviews with a specific and reported search strategy and/or meta-analyses (rapid reviews were also included due to their necessity/value during a pandemic); (2) full-text articles in English; (3) studies on characteristics and outcomes of pregnant or recently pregnant (postpartum, post-abortion, post-miscarriage) women with RT-PCR confirmed or suspected (based on clinical and imaging findings) SARS-CoV-2 infection, characteristics and outcomes of their neonates or the potential of SARS-CoV-2 vertical transmission; (4) neonates whose mothers were confirmed or suspected as being infected with SARS-CoV-2.

Exclusion criteria were as follows: (1) non-reviews, protocols and reviews without a systematic search strategy and/or synthesis of data; (2) animal or in vitro studies; (3) studies including neonates other than those whose mothers had a confirmed or suspected COVID-19 (e.g., neonates just being admitted to hospital due to suspected pneumonia of unknown origin).

### 2.2. Search Strategy and Study Selection

The search strategy followed the Preferred Reporting Items for Systematic Review and Meta-analysis (PRISMA) guideline [6]. Three electronic databases, namely, PubMed, Scopus and the Cochrane Database of Systematic Reviews, were searched. Keywords employed were (COVID-19 OR SARS-CoV-2 OR “Coronavirus disease 2019”) AND (“Neonatal outcom*” OR “Neonatal characteristic*” OR “Maternal outcom*” OR “maternal characteristic*” OR “pregnancy outcom*” OR “vertical transmission”) (Appendix A). The last search took place on 10 September 2020. A snow-ball procedure was also implemented by hand-searching the reference lists of included systematic reviews for additional sources. All retrieved studies were imported into the Rayyan QCRI and duplicated articles were removed. Two independent researchers (CP and ER) initially screened all articles based on title and abstract, categorizing them as “included”, “excluded” or “maybe”. Any disagreements or “maybes” were resolved by consensus, along with the involvement of a third reviewer (MF). Only totally irrelevant articles were excluded at this stage. Subsequently, full text assessment of the included studies was performed by the same two authors (CP and ER) to determine study eligibility based on the inclusion and exclusion criteria. Evolving inconsistencies were again resolved by a third author (MF). Articles that did not fulfill the agreed eligibility criteria were removed.

### 2.3. Data Extraction

Data from the eligible studies were independently extracted by two authors (MP and MF) using the predefined standardized extraction form, and then verified by a third author (MP). Extracted variables from each systematic review are extensively presented in our registered protocol (PROSPERO record: CRD42020214447) [7].

### 2.4. Outcomes

#### 2.4.1. Primary Outcomes

Our primary outcomes included: (1) mode of delivery (vaginal, either spontaneous or operative, and cesarean section, either emergency or elective); (2) preterm delivery (birth < 37 weeks) and preterm labor (labor onset <37 weeks of gestation) [8]; (3) premature rupture of membranes (PROM), defined as rupture occurring before labor onset and preterm premature rupture of membranes (pPROM), defined as rupture occurring before 37 weeks of gestation [9]; (4) abortions and miscarriages.

#### 2.4.2. Secondary Outcomes

Secondary outcomes are divided into maternal COVID-19-related, fetal/neonatal and data on SARS-CoV-2 vertical transmission potential.

Maternal COVID-19-related outcomes included: (1) clinical symptoms (i.e., fever, cough, myalgia, fatigue or weakness, dyspnea, shortness of breath (SOB), sore throat, headache, diarrhea and loss of smell and/or taste); (2) laboratory measurements (i.e., white blood cell count, thrombocytopenia, inflammatory markers such as C-Reactive Protein (CRP) and procalcitonin (PCT), D-dimers and liver function tests (LFTs)) and (3) imaging findings; (4) maternal ICU admission; (5) need for mechanical ventilation (either invasive or non-invasive) and (6) maternal mortality.

Fetal and neonatal outcomes included: (7) fetal distress, as defined by the authors of each study; (8) Fetal Growth Restriction (FGR); (9) stillbirth; and (10) low birth weight (<2500 g); (11) Apgar scores at 1 and 5 min; (12) neonatal asphyxia; (13) neonatal admission to the ICU (NICU) and (14) neonatal mortality (not including stillbirths).

Concerning vertical transmission potential, outcomes included: (15) PCR positive neonates; (16) SARS-CoV-2 IgM (+) and/or IgG (+) neonates and (17) types (and, when provided numbers) of samples collected to test for mother-to-child transmission potential (other than nasopharyngeal, oropharyngeal or “throat” swabs and neonatal blood serum).

### 2.5. Quality Assessment

The methodological quality of the systematic reviews was assessed by the use of the Assessment of Multiple Systematic Reviews 2 (AMSTAR 2) checklist by two independent reviewers (CP and ER) [10]. Any discrepancies arising during the evaluation process were resolved through consensus and arbitrated by a third author (MP).

### 2.6. Data Synthesis and Presentation

Categorical variables are presented as frequencies and percentages. Numerical variables are presented as mean, standard deviation (SD). In this case, variables were provided as median, range or median, interquartile range, they were transformed into mean, SD using the methods described by Hozo et al. and the Cochrane handbook [11,12]. We mainly presented rates of our outcomes of interest as ranges. When calculating ranges, only clearly reported raw rates or raw rates calculable by clearly provided data (i.e., exact number of subjects with the outcome of interest (n) and exact number of subjects screened for the outcome (N)) were taken into account, since dividing affected subjects by the whole included population would probably underestimate the effect. When calculating ranges of raw rates, reported pooled proportions were not taken into account. In order to portray currently available data more objectively, we also presented data from the largest samples on the analyzed outcome. In the case of meta-analyses, we extracted pooled outcomes along with their 95% confidence intervals and Higgins I^2^ statistics [13]. The same data were also presented after excluding “very low” or “low quality” studies and after taking into account only “moderate” and “high quality” studies, as these resulted from the quality assessment using the AMSTAR-2 tool. In this second analysis, studies of “moderate quality” were also included, taking into consideration that the majority of available systematic reviews were primarily based on preliminary data and were also designed and conducted at a time at which urgent answers were needed.

## 3. Results

The initial literature search yielded 750 studies. After removal of duplicates, 515 articles were screened based on title and abstract only, with 444 of them being excluded. Full texts were screened from 71 studies. Of these, 32 were excluded. The detailed study selection flowchart is presented in Figure 1.

### 3.1. Study Characteristics

After the application of the eligibility criteria, 39 studies were finally included [14,15,16,17,18,19,20,21,22,23,24,25,26,27,28,29,30,31,32,33,34,35,36,37,38,39,40,41,42,43,44,45,46,47,48,49,50,51,52], one (2.6%) of which was described as “rapid” [42] and two (5.1%) as “scoping” systematic reviews [30,50]. Thirteen (33%) of them also undertook a meta-analysis [14,16,18,19,24,25,27,36,38,40,48,49,52]. Twenty-nine (74%) reviews evaluated quality of individual studies [14,15,16,19,21,22,23,24,25,26,27,29,30,31,32,33,34,35,37,41,42,43,44,46,47,48,51,52], with one of them doing so, in a subjective manner, without utilizing any assessment tool [42]. Among the eligible systematic reviews’ last search dates, the most recent one took place on 8 July 2020. Nineteen studies (48.7%) contained only confirmed women for SARS-CoV-2 [14,15,16,18,19,20,23,24,27,28,36,37,38,43,45,46,48,49,52] (2 studies used the term “laboratory confirmed COVID-19” [14,19], 6 studies the term “confirmed COVID-19” [16,18,28,38,48,49], 10 studies the term “PCR-confirmed” [15,20,23,24,27,36,37,43,46,52] and one study the term “COVID-19 positive” [45]), fourteen (35.9%) included either PCR-confirmed or women whose diagnosis of COVID-19 was based on clinical and/or radiological findings, without further laboratory confirmation [17,21,29,30,31,32,33,34,35,40,44,47,50,51], while six (15.4%) did not report exactly whether COVID-19 was suspected or confirmed in included participants [22,25,26,39,41,42]. Three reviews also provided the impact of prior coronaviruses on pregnancy [14,20,48], one included neonates of women with COVID-19 without reporting on their mothers and pregnancy [37], two referred to neonates born to COVID-19 positive mothers, also conducting reviews on pediatric COVID-19 [38,46], while one review included only pregnant women admitted to ICU, focusing on calculating their case fatality ratio [18].

Evidence on participants’ trimester of SARS-CoV-2 infection could be extracted by 16 reviews [20,23,27,29,30,34,35,36,41,42,43,44,46,47,51,52]. Only two reviews (12.5%) clearly stated the inclusion of first-trimester pregnancies with SARS-CoV-2 infection, with rates of women that were infected during their first trimester of pregnancy in these two reviews being 5% and 6% [23,34]. Seven reviews (46.7%) clearly stated the inclusion of second trimester pregnancies with SARS-CoV-2 infection, with rates of women that were infected during their second trimester varying between 1% and 10% [20,23,34,35,36,42,44]. Five reviews (31.25%) clearly included only second and third trimester pregnancies [20,35,36,42,44]; three reviews (18.75%) stated including mostly third trimester pregnancies, without further clarifying the number of pregnancies in first or second trimester [27,43,51], and six reviews (37.5%) analyzed exclusively third trimester pregnancies [29,30,41,46,47,52].

### 3.2. Quality Assessment

Twelve reviews (30.8%) were found to be of “very low quality” [16,17,18,20,28,33,34,38,39,42,50,51], 11 (28.2%) of “low quality” [14,15,19,21,22,23,29,30,37,41,44], 13 (33.3%) of “moderate” [24,26,31,32,35,36,40,43,45,46,47,49,52], and three (7.7%) of “high quality” [25,27,48] (Appendix A).

### 3.3. Primary Outcomes

#### 3.3.1. Mode of Delivery

Thirty-five reviews exhibited data on selected delivery modes for both preterm and term gestations [14,15,16,17,19,20,21,22,23,24,25,26,27,28,29,30,31,32,33,34,35,36,38,39,40,41,42,43,44,46,47,48,50,51,52]. Reported cesarean section (CS) rates ranged between 52.3% (390/746) and 95.8% (46/48) [14,15,16,17,19,20,21,22,23,24,25,26,27,28,29,30,31,32,33,34,35,36,38,40,41,42,43,44,46,47,48,50,51,52]. The review with the largest number of included deliveries found a CS rate of 54.8% (1060/1933; pooled proportion: 64.7% (56.5–72.6, I^2^ = 91.3%)) [40]. In contrast, vaginal delivery rates ranged between 4.2% (2/48) and 44.7% (856/1916 (largest sample); pooled: 35% (27–43, I^2^ = 91.4%)) [14,16,20,22,23,26,28,29,30,31,32,33,34,35,36,38,39,40,41,42,43,44,47,50,51,52]. After including only moderate and high-quality studies in a separate analysis, both the ranges of reported CS rates (52.3–94%) [24,25,26,27,31,32,35,36,40,43,46,47,48,52] and vaginal delivery rates (6–44.7%) [26,31,32,35,36,40,43,46,47,48,52] slightly changed.

Eight reviews reported on COVID-19-related selection of a specific delivery mode [20,23,27,31,34,39,44,47]. Muhidin et al. described two vaginal deliveries that were decided due to the absence of respiratory symptoms in positive COVID-19 mothers [47]. Rates of CS, whose primary indication was COVID-19, varied between 7.7% (4/59) and 60.4% (218/361) [20,23,27,31,34,39,44]. In particular, one review found that maternal SARS-CoV-2 infection was the primary indication for 49.6% (59/119) of preterm cesarean deliveries and 65.7% (159/242) of term cesarean deliveries [23]. Khalil et al., including the largest sample for this outcome, calculated 19.1% [95/497; pooled proportion: 19% (8.9-36.6, I^2^ = 89.4%)] of deliveries, which were decided based on COVID-19–associated parameters [27]. After accounting for quality of studies, no range could be extracted on rates of CS, whose primary indication was COVID-19.

#### 3.3.2. Preterm Delivery and Preterm Labor

In 32 reviews [14,15,16,17,20,21,22,23,24,25,26,27,28,29,30,32,34,35,36,38,39,40,41,42,43,44,45,46,48,50,51,52], reported rates of preterm deliveries varied between 14.3% (86/602) and 63.8% (30/47). In the largest sample, with 1872 included deliveries, preterm delivery rate was 20.6% (pooled: 16.9 (13.2–20.9, I^2^ = 71.5%) [40]. Two studies also gave spontaneous preterm birth rates of 5% (22/440; pooled: 5% (3.3–7.5, I^2^ = 82.3%) and 6.4% (56/870; pooled: 6.1% (3.4–9.4, I^2^ = 55.0%) [27,40], while another review found a medically-indicated preterm birth rate of 21.4% (92/430; pooled: 18.4% (8.3–35.8, I^2^ = 87.4%) [27]. Finally, as mentioned above, Turan et al. found that 49.6% (59/119) of preterm deliveries had maternal SARS-CoV-2 infection as their primary indication [23]. After extracting rates from only the moderate and high-quality studies, only the range of preterm delivery rates changed (20.6–63.8%) [24,25,26,27,32,35,36,40,43,45,46,48,52].

Nine reviews presented data on “preterm labor” [19,23,28,29,32,39,41,44,47], with one of them providing a composite outcome of pPROM and preterm labor [32], another one specifically focusing on spontaneous preterm labor (3.9% (25/637)) [23] and Della Gatta et al. mentioning them as part of indications for CS (4(11.8%)). With regard to the rest of the reviews, reported preterm labor rates ranged between 22.7% (22/97) and 32.2%. After accounting for quality of studies, no range on preterm labor rates could be extracted.

#### 3.3.3. PROM and pPROM

Twelve reviews provided data on PROM [22,25,26,28,29,30,38,39,41,44,47,52], among which one review extracted their rate based on the number of included studies (not participants) that reported PROM (5 studies (38.5%)) [25] and two reviews extracted their rates from the number of women with comorbidities (7(16%) [30], or among indications for CS (9(26.5%)) [44] and not the entire included population. Analysis from the rest of the reviews led to a range between 5.3% (5/95) and 12.7% (16/126) (which represented the largest sample analyzed for this outcome) [28]. After the exclusion of very low and low-quality studies, no range on PROM rates could be extracted, while a rate of 5.3% represented the largest sample analyzed for the outcome [26].

Six reviews estimated pPROM rates [14,20,23,32,40,48]. Among them, one presented pPROM as part of a composite outcome, which also included preterm labor [32]. The extraction of data from the remaining studies resulted in a range between 6.4% ((28/436, largest sample); pooled: 5% (3–8, I^2^ = 45.6%)) and 16.1% (5/31) [40]. After considering only the provided rates by moderate and high-quality studies [32,40,48], no differences with the aforementioned results were found.

One review did not elucidate the type of membrane rupture [36].

#### 3.3.4. Miscarriages and Abortions

Ten articles dealt with miscarriages [14,17,23,26,31,34,41,42,45,50]. One review integrated them into pregnancy terminations (1.4% (4/295)) [31], one into intrauterine fetal deaths (1(3%)) [42], while another one described them as “spontaneous abortions” (0.8% (3/385)) [50]. Taking into account reviews that calculated these rates for their entire included population, miscarriage rates were <2.5%. However, a review with a total sample of 637 participants, reported rates of 16.1% (5/31) and 3.6% (2/55) for first and second trimester infections only (totally: 8.1% (7/87)), respectively [23]. The reported rates by moderate and high-quality studies were ≤2% [26,31,45].

Six reviews calculated abortions [23,24,31,32,41,50], with one of them also studying the composite outcome of pregnancy terminations and miscarriages [31]. The maximum reported the absolute number of abortions was 9 in a cohort of 87 first and second trimester infections [23]; in all cases, decisions were taken due to anxiety for potential COVID-19-related pregnancy adverse outcomes [23]. Interestingly, in a total sample of 385 women with confirmed/suspected COVID-19, another review reported four (1%) “technically induced abortions” and three (0.8%) were characterized as “spontaneous” (i.e., the aforementioned miscarriages) [50]. Finally, five early terminations and three “threatened abortions” were described among 790 pregnant women [24]. (Nonetheless, the term “threatened abortions” presumably relates to “threatened miscarriages”.)

Table 1 summarizes the included systematic reviews’ results with regard to the primary outcomes before and after the exclusion of very low and low-quality studies, respectively.

### 3.4. Secondary Outcomes

#### 3.4.1. Symptoms

Thirty studies [14,16,17,19,20,21,22,23,24,25,26,27,28,29,30,31,32,34,35,36,40,41,43,44,45,46,47,48,50,52] reported on symptoms of infected mothers. Table 2 presents data on maternal symptoms provided by the included systematic reviews before and after the exclusion of very low and low-quality studies, respectively.

#### 3.4.2. Laboratory Findings

Twenty-three reviews [14,19,20,21,23,24,25,27,28,29,31,32,34,35,36,40,43,44,47,48,49,50,52] focused on laboratory findings of pregnant women. Table 3 presents data on maternal laboratory parameters provided by the included systematic reviews before and after the exclusion of very low and low-quality studies, respectively. Abbreviations used in all tables are also summarized and explained in Appendix A.

#### 3.4.3. Radiological Imaging Findings

Seventeen reviews emphasized the imaging findings observed in women, with high heterogeneity in the way of their reporting between the different articles [14,19,20,23,28,29,31,32,34,35,40,42,44,45,47,50,52]. Diriba et al. provided information regarding the whole coronavirus spectrum [14]. CT abnormality rates ranged between 30.4% (599/1968) and 98.7% (78/79) among the different studies. The latter remained exactly the same after taking into account only studies of moderate and high quality [31,32,35,40,45,47,52]. The most commonly observed radiological pattern was ground glass opacities with rates from 41.5% (27/65) to 81.6% (102/125) and a rate of 63.6% (246/387; pooled: 68.6% (40.8–91, I^2^ = 96%)) representing the largest sample analyzed [40]. After accounting for quality, a range could no longer be extracted. Less common patterns included “patchy infiltrates/shadowing”, “patchy consolidation” and “reticular” pattern. Findings were primarily observed bilaterally.

#### 3.4.4. ICU Admission

Twenty-three reviews [14,18,20,21,23,25,27,28,29,31,32,35,36,39,40,41,42,43,44,47,48,50,51] reported on women’s admissions to the ICU. One article entirely included women admitted to the ICU, with the aim of calculating their case fatality ratio [18]. Admission rates were noticed to be between 3% (323/10901 (largest sample); pooled: 4.1 (1.8–7.1, I^2^ = 93.6%)) and 28.5% (53/186). After accounting for quality of studies, ICU admission rates ranged between 3% and 10%, with 3% still representing the largest sample [25,27,31,32,35,36,40,43,47,48].

#### 3.4.5. Mechanical Ventilation

Mechanical ventilation rates ranged between 1.4% (155/10713) and 12% (5/41), (18 reviews) [20,22,23,27,28,29,31,32,35,40,42,44,45,47,48,50,51,52]. Turan et al. actually provided a rate of mechanical ventilations in patients admitted to the ICU (83.6% (51/61)). The rates of 1.4% (155/10713; pooled: 2.6% (0.8–5.2, I^2^ = 93.5%)) and 5.5% (92/1680; pooled: 3.4% (1.5–7.7, I^2^ = 90.2%) represented the largest samples analyzed and both regarded invasive mechanical ventilation [27,40]. After accounting for quality, mechanical ventilation rates varied between 1.4% and 5.5%, with 1.4% still representing the largest sample [27,31,32,35,40,45,47,48,52].

#### 3.4.6. Maternal Death

Taking into account 27 relevant reviews, maternal death rates did not exceed 2% [14,17,18,19,20,21,22,23,25,27,28,29,30,31,32,34,35,36,40,42,43,45,47,48,50,51,52]. Among them, Kim et al. calculated a case-fatality ratio of 12.9%, including only women in the ICU [18], while Gao et al. analyzed a pooled composite outcome of severe case or death (pooled: 12% (3–20, I^2^  =  0%)) [19]. Indicatively, the two largest reviews with regard to maternal deaths presented rates of 1.7% (43/2468; pooled 0.9% (0.4–2.3, I^2^ = 73.4%)) [27] and 0.6% (73/11580; pooled: 0.1% (0–0.7, I^2^ = 80.2%)) [40]. After the exclusion of very low and low-quality studies, maternal death rates remained <2%, with 0.6% still representing the largest sample [25,27,31,32,35,40,43,45,47,48,52].

#### 3.4.7. Fetal Distress

Twenty reviews [14,19,20,22,23,26,28,29,30,35,36,38,40,41,44,45,47,48,50,52] gathered information on distress of included fetuses. Rates varied between 7.8% (20 fetal distresses) and 61.1% (11/18), with the largest sample corresponding to a rate of 8.5% (25/293; pooled: 8% (5–12, I^2^ = 0%)) [40]. When only moderate and high-quality studies were taken into account, rates varied between 8.5% and 61.1%, with the rate 8.5% corresponding to the largest sample [26,35,36,40,45,47,48,52].

#### 3.4.8. Fetal/Intrauterine Growth Restriction (FGR)

Data on FGR are limited with only three reviews focusing on this outcome [14,45,48]. Reported rates were 0% (pooled: 0% (0–22, I^2^ = 0%), 1.2% (1/86, 0–6.3) and 9% (corresponding to just 1 case of FGR).

#### 3.4.9. Stillbirth

Reported stillbirth rates of 26 reviews did not exceed 2.5% [16,17,19,20,23,24,25,26,27,28,29,32,34,35,36,40,41,42,43,44,45,47,48,50,51,52]. Four reviews presented a composite outcome of stillbirths and neonatal deaths [16,24,48,50]. Gao et al. chose to present stillbirths along with neonatal asphyxia and death as a single outcome (9% (3–21, I^2^ = 0%)) [19]. The two largest collected series of deliveries analyzed for this purpose resulted in stillbirth rates of 0.9% (12/1362; pooled: 0.9 (0.5–1.5, I^2^ = 0%)) and 0.6% (18/2837; pooled: 0% (0–0.1, I^2^ = 0%) [27,40]. By taking into account only the rates provided by moderate and high-quality reviews, the rates were found to vary between 0.6% and 2.4%, with the 0.6% representing the largest sample [24,25,26,27,32,35,36,40,43,45,47,48,52].

#### 3.4.10. Low Birth Weight

According to data acquired from 15 reviews [15,24,25,26,28,29,31,34,35,39,41,46,47,50,51], neonatal low birth weight rates ranged between 5.3% (equivalent to four neonates) and 47.4% (9/19). Rates arising from the two largest samples’ analysis were 6% and (28/493) and 7.8% (20/256) [34,50]. After accounting for quality of studies, a slight difference in range was observed (7.8–47.4%) and the rate of 7.8% was found to represent the largest sample [24,25,26,31,35,46,47].

#### 3.4.11. Apgar Scores

Twenty-three reviews reported on neonatal Apgar scores with high heterogeneity in presentation of findings [14,17,21,23,24,25,26,28,29,30,31,32,35,36,38,39,40,45,46,47,48,51,52]. Six reviews reported no neonates with scores <7 at 1 and 5 min [26,29,31,35,38,46]. A meta-analysis estimated mean Apgar scores of 8.811 (8.382–9.240, I^2^ = 88.9%) and 9.516 (9.136–9.895, I^2^ = 82.9%) at 1 and 5 min, respectively [52]. Two reviews reported six neonates with scores < 7, all following preterm delivery due to fetal distress in mothers with critical COVID-19 [17,23]. Finally, Allotey et al., which included the largest number of neonates for this outcome, calculated 2.2% (11/500) of neonates with “abnormal” Apgar scores at 5 min [40].

#### 3.4.12. Neonatal Asphyxia

Twelve reviews included data on neonatal asphyxia [14,17,19,24,26,28,31,32,41,48,51,52] with one composite with stillbirth and neonatal death as a single outcome [19]. The maximum reported rate was 13% and was equivalent to two incidents, one of which being observed in a SARS-CoV-2 positive neonate [32]. Yoon et al. found one neonatal asphyxia among four SARS-CoV-2 positive neonates [28]. Finally, the two largest series exhibited rates of 1.8% (3/168) and 0.6% (1/161) [28,31]. After accounting for quality, neonatal asphyxia rates remained ≤13%, with 0.6% representing the largest sample [24,26,31,32,48,52].

#### 3.4.13. NICU Admission

Data retrieved from 12 reviews led to a range of NICU admission rates of 3.1% (8 admissions)–76.9% (11/13) [14,15,21,23,31,35,40,43,44,48,50,52]. However, Dhir et al. calculated admission rate of only SARS-CoV-2 positive neonates (38% (22/58)) [15]. Three large series demonstrated rates of 64.9% (137/211) [21], 11.3% (54/479) (and 52 (96.3%) of these admissions being preterm neonates and SARS-CoV-2 negative) [23] and 27.3% (368/1348; pooled: 24.6% (14.3–36.6, I^2^ = 94.9%)) [40]. After accounting for quality of studies, NICU admission rates varied between 10% and 76.9%, with 27.3% still representing the largest sample [31,35,40,43,48,52].

#### 3.4.14. Neonatal Mortality

Thirty-five reviews reported on neonatal mortality [14,15,16,17,19,20,21,22,23,24,25,26,27,28,29,30,31,32,34,35,36,38,39,40,41,42,43,44,45,46,47,48,50,51,52] with four of them describing a composite outcome of stillbirth and neonatal death [16,24,48,50] and one an outcome of neonatal death, stillbirth and neonatal asphyxia [19]. Based on the remaining 27 studies, neonatal mortality did not exceed 3% (1/29). A rate of 0.3% (6/1728) represented the largest sample [40], while the maximum absolute number of neonatal deaths was 10 [34]. No deaths were directly associated with neonatal SARS-CoV-2 infection. After the exclusion of studies of very low and low quality, neonatal mortality did not exceed 2.4% with the rate of 0.3% still representing the largest sample [24,25,26,27,31,32,35,36,40,43,45,46,47,48,52].

#### 3.4.15. Neonatal PCR Positivity

Twenty-eight reviews found neonates that were PCR positive to SARS-CoV-2 [15,16,17,19,20,23,24,25,26,27,28,29,30,31,32,33,34,35,37,38,39,41,43,44,45,50,51,52]. Neonatal PCR positivity rates ranged between 1.6% (4/256) and 10% (7/68). It should be noted that Dubey et al. found a 14% (6/43) neonatal positivity rate, when conducting a subgroup analysis of only individual case studies [24]. The two largest samples, regarding neonatal PCR positivity, gave rates of 2.5% (19/751; pooled: 1.4% (0.4–4.7, I^2^ = 59.8%) and 5.5% (58/1048) [15,27]. After taking into account only studies of moderate or high quality, neonatal PCR positivity rates ranged from 2% to 7%, with the rate of 2.5% representing the largest sample [24,25,26,27,31,32,35,43,45,52].

#### 3.4.16. Neonatal Serum Antibody Positivity

Twelve reviews found serum antibody positive (SARS-CoV-2 IgM and/or IgG) neonates [16,17,20,28,32,34,37,39,41,43,45,50], with one of them noteworthily reporting three neonates with IgM positivity, yet negative PCR of nasopharyngeal swabs [43].

#### 3.4.17. Samples Tested for SARS-CoV-2

Data on types and/or number of collected samples to be examined in view of vertical transmission potential derived from 22 reviews [14,17,20,22,23,27,28,29,30,31,32,33,35,36,37,38,41,45,47,50,51,52]. Collected samples included placenta surface swabs, amniotic fluid, umbilical cord blood, neonatal gastric juice, urine, stool and anal swabs as well as maternal vaginal secretions. Only five reviews reported positive samples [17,20,22,23,29]; one found four positive placenta and one positive cord blood samples, paradoxically all accompanied by negative neonatal nasopharyngeal swabs [17], two of them reported one positive amniotic fluid (each accompanied by negative placenta and cord blood samples) [20,29], one found 3 positive among 11 placental or membrane swabs [22], and the last one reported one positive placental surface swab among a total of six placentas [23].

## 4. Discussion

In this overview of systematic reviews, we tried to describe the obstetric-perinatal and neonatal outcome of infected pregnant women and their newborns during the SARS-COV-2 pandemic. After application of our inclusion criteria, we analyzed 39 reviews. CS rates were between 52.3 and 95.8%, preterm deliveries were between 14.3 and 63.8%, PROM between 5.3 and 12.7% and pPROM between 6.4 and 16.1%. Fever and cough were the most frequently recorded symptoms, while maternal ICU admission and mechanical ventilation rates were high. Neonatal ICU admission and mortality rates were 3.1–76.9% and <3%, respectively. Neonatal PCR positivity rates ranged between 1.6% and 10%. The methodological quality of the reviews was assessed through the AMSTAR 2 checklist; interestingly, almost 2/3 of the reviews were assessed as being at “very low” or “low” quality, while only 7% of them were judged as being at “high” quality. After taking into account only the rates provided by moderate and high-quality studies, CS rates, preterm deliveries and pPROM were similar; fever and cough were again the most frequently recorded symptoms, while maternal ICU admission and mechanical ventilation rates were found somewhat lower than the primary analysis. Similarly, all the rest of the secondary outcomes remained unchanged.

In general, the symptomatic infection in pregnant women seems to be of lower incidence compared to the general population. Nonetheless, in cases of pregnant women with symptoms like fever and cough, negative outcomes may be expected especially after hospitalization due to the severity of the symptoms required.

Concerning the mode of delivery, selection of CS was reported at the extremely high levels of 52.3–95.8%, although both the International Federation of Gynecology and Obstetrics and the Royal College of Obstetricians and Gynecologists recommended that this should not be influenced by the COVID-19 status [53,54]. This is probably linked with the lack of formal recommendations at the onset of the pandemic. Interestingly, Dubey et al. showed a further higher incidence of CS in Chinese compared to US and European studies [24], which was probably due to the already high baseline rates (41.5%) and the differences in local practices in China [35,55]. In addition, higher incidences were observed by the first reviews published, while cohort sizes were small, mainly coming from China, and including reports with high publication bias [56]; in contrast, one review demonstrated no difference in the CS rates between pregnant and recently pregnant women with COVID-19 compared to those without [40]. Furthermore, details on COVID-19 as an indication for elective CS were not broadly reported, with the reasons mentioned to include worsening of the maternal condition, need for antiviral treatment/facilitation of infection control procedures, relief of abdominal pressure for better respiration and minimization of the transmission potential to the neonate [29,32,34,39]; of these, the first two constituted the main reasons [20,23,27,34,39]. There were, however, cases in which the fear of neonatal infection was the only reason stated [57,58], although one review doubted this fear, discovering that only 2.7% (8/292) of vaginally delivered neonates were found positive, compared to 5.3% (20/374) of those born through CS [44].

The majority of the included studies examined preterm delivery rates. We noticed a wide range of reported proportions, primarily explained by the heterogeneity of the sample sizes. Combining studies with large samples, we observed a range of 20.6–25% [15,40], which is higher than the worldwide baseline [59]. Also, the comparison of pregnant and recently pregnant women with, and without the disease, exhibited greater odds of preterm birth in the former group [40]. Interestingly, the increase in preterm births was not portrayed by spontaneous preterm birth rates, which were relatively low (5–6%) and comparable with those of the general population [23,40,60]. Khalil et al. reported a medically indicated preterm birth rate of 18% that was proximate to their estimated proportion of overall preterm births [27]. In another review, however, half of preterm births involved cases of fetal or maternal compromise [23]. Thus, safe conclusions on the increase of preterm births cannot still be extracted, yet parameters related to medical management might be involved. Despite that preterm labor rates resemble those of preterm birth, data on preterm labor should be interpreted with extreme caution due to the potential misuse of the terms “preterm birth” and “preterm labor”. Gao et al. attributed the elevated rates of preterm labor to induction of labor [19].

Available data on the rest of our primary outcomes were less adequate. Only 11 studies out of 39 investigated PROMs, all of them with small populations, leading to differential results. After analyzing a sample of 126 women, Yoon et al. provided a rate of 12.7% [28], only a little higher than the 5–10% rates of the general population [61]. Similar data limitations apply to pPROM. The largest sample for this outcome (n = 436) demonstrated a proportion of 6.4%, which is moderately higher than its prevalence in the general population (3%) [40,61]. Since all reviews included few or no women in the 1st and 2nd trimester of pregnancy, scarce data were available on miscarriages and abortions. It is possible that asymptomatic cases in this period are underreported due to fewer tests performed and poorer obstetrical surveillance [31,34]. Moreover, taking into account the high sequence similarity of SARS-CoV-2 and SARS-CoV infections [62] and the excessive miscarriage rates reported in pregnant women with the latter [14,28], an association of miscarriages with the former cannot be excluded. Notably, one of our included reviews found an 8.1% rate of miscarriages in 86 first and second trimester pregnancies [23]. In addition, a case of miscarriage during the second trimester of pregnancy, in which SARS-CoV-2 was detected in the placenta, should raise further concerns [63]. Among the terminations of pregnancies reported, while few are witnessed in cases with severe complications, most of them were linked to psychological factors [24,64]. For instance, Turan et al. described nine cases of induced abortions that were related to concerns on the fetal development after maternal infection [23].

Asymptomatic women ranged between 7.5% and 32.6%; of note, this percentage might be higher, given that pregnant women undergo a larger number of tests than the general population [40]. It is known -and recently published- that SARS-CoV-2 has a prolonged and nonspecific disease course during pregnancy and at six weeks postpartum [65]. Among symptoms, the most commonly reported were fever and cough. This sequence highly suggests that clinical symptoms of pregnant might resemble those of non-pregnant women. However, differences in symptoms’ frequencies between the two groups might exist, with common symptoms such as fever, cough and myalgia being less frequently reported by pregnant women compared to non-pregnant [40,66]. Furthermore, according to Dubey et al., certain symptoms at presentation, such as myalgia, might be associated with adverse pregnancy outcomes [24]. Differences of systematic reviews in reported symptom rates may be explained by the fact that preliminary reviews included studies primarily based on symptom-based screening, whereas subsequent reviews included studies with more participants diagnosed during random screening processes. Reported maternal ICU admission, mechanical ventilation and mortality rates were high, when compared with non-pregnant women [40,66]: a possible interpretation to this finding was the existence of other comorbidities accompanying pregnancy, compared with non-pregnant women at a similar age [27]. Concerning mortality rates, available data were more contradictory [40,66]. The increased rates, when reported, have been also attributed to the specific healthcare provision of the participants’ countries [18,20]. Markedly, the updated CDC report, which demonstrated a greater risk of death for pregnant women, analyzed only symptomatic women [66].

Among the adverse neonatal or fetal outcomes, abnormal APGAR scores, neonatal asphyxia, stillbirth and neonatal death rates have been found similar with uninfected fetuses, and when also analyzing the results of reviews with larger samples [40]; in contrast, there was only a higher risk of NICU admission. According to Juan et al., precautionary investigation and monitoring due to their mothers’ infection was the main reason for this [31]. As for the neonatal morbidity, it was higher and linked with NICU admission and preterm birth in mothers with severe or critical infection [23]; of note, none of the 54 reported neonates were infected by SARS-CoV-2 [23]. It is evident that both hypoxemia and respiratory failure of severely affected mothers can cause pre-placental hypoxia, which leads to fetal distress, preterm labor and stillbirth, not only for the offspring, but also for maternal ventilation purposes [28,67]. In conclusion, it seems that prematurity and severe maternal disease are the primary contributors to the reported high neonatal morbidity and mortality. To our knowledge, no death has been reported among SARS-CoV-2 positive neonates up until the time of writing. As for prematurity, this was due to maternal or fetal malperfusion findings of placental pathology, potentially predisposing the fetus to chronic hypoxia [27,67]. This led to failure of the fetus to thrive to its genetically determined growth, which is associated with distress, asphyxia and increased perinatal mortality [67].

Data from analyzed systematic reviews could not lead to definite conclusions on vertical transmission potential. Even in RT-PCR positive neonates, who also had elevated serum anti-SARS-CoV-2 IgM and IgG antibodies, amniotic fluid, placenta, and umbilical cord blood samples were negative [17,28]. Therefore, vertical transmission, although highly anticipated in such cases, was not confirmed. After studying 58 neonatal infections, Dhir et al. stated that 41 of them were probably transmitted postpartum, 0 intrapartum and 13 could not be assigned to a transmission mode [15]. However, considering cases of PCR positive placenta, amniotic fluid or cord blood samples, vertical transmission cannot be excluded [17,20,22,23,29].

Finally, it should not be overseen that assessment of the existing systematic reviews, using the AMSTAR 2 tool, classified 59% of them as “critically low quality” to “low quality”. However, it should be taken into account that currently available systematic reviews based their results on a large number of preliminary studies conducted due to the urgent need for quick answers. For these reasons, further original studies with women from all pregnancy trimesters and longer follow-up periods, and, consequently, further systematic reviews synthesizing their results, are required to provide clinical practice with more definite answers regarding the effects of SARS-CoV-2 infection on such a sensitive group of patients as pregnant women and their fetuses/neonates. So far, data are not robust enough to lead to definite points and regulations. The SARS-CoV-2 pandemic impact on maternity care all over the world has yielded various responses, including labor care and choice of place of birth, increased risk of adverse decisions, reduction in antenatal and postnatal ‘face to face’ care provision and various necessary adjustments with unknown impact on women and offspring’ wellbeing, or on women’s experiences of birth [68].

### Limitations and Strengths

This is the first overview summarizing all existing data on pregnancy, maternal and neonatal characteristics and outcomes in a systematic manner on SARS-CoV-2 infection. We extracted data from systematic reviews, which are considered high-quality evidence. We assessed both quantity and quality of evidence on each outcome and emphasized results from large samples to increase objectivity of narrative reporting as much as possible. We also conducted a separate analysis, extracting results only from moderate and high-quality studies. Adversely, our overview is subject to certain limitations. Results are presented in a narrative way using ranges as the primary mean of quantification. We also included studies with both RT-PCR positive women and women with suspected infection based on their clinical and imaging manifestations, whereas, if excluding them, we might have missed a considerable source of information. Finally, we noted very wide confidence intervals and considerable heterogeneity (I^2^ > 70%) of the included studies, which are both linked with a high uncertainty of the evidence through true effect/result. The reasons include various (usually small) sample sizes and characteristics of the populations studied, the number of studies combined and low precision of the individual study estimates. Thus, the results of the current study should be interpreted with caution, something that is sensible, due to the ongoing nature of the disease itself and the premature conduction of studies.

## 5. Conclusions

In conclusion, a rapid increase of CS was observed, especially at the beginning of the pandemic, most likely due to lack of knowledge and robust recommendations. Preterm birth rates were elevated, with iatrogenic reasons potentially involved. While pregnant women have an increased risk for ICU admission and mechanical ventilation, the mortality trend was not clearly elucidated. Neonates were more frequently admitted to NICU, which may be attributed to precaution or severe maternal infection. The remainder of the fetal/neonatal outcomes presented are of low incidence and were possibly related to prematurity. Even though neonatal infections were rare, the probability of vertical transmission cannot be eliminated. After taking into account only moderate and high-quality studies, ranges of our primary outcomes remained unchanged, while among our secondary outcomes, maternal ICU admission and mechanical ventilation rates were found relatively lower. Further original studies with women from all trimesters and longer follow-up periods are needed.

## Figures and Tables

**Figure 1 ijerph-18-00596-f001:**
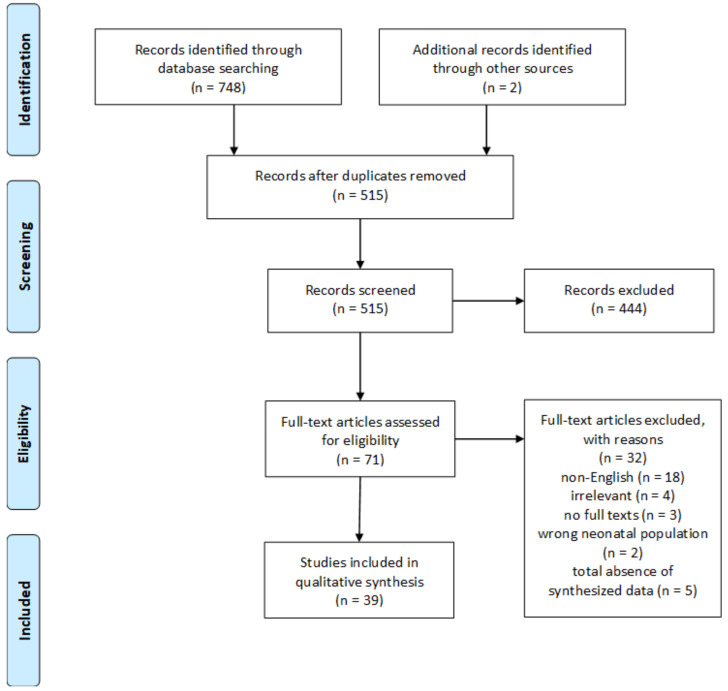
Flow diagram of the overview.

**Table 1 ijerph-18-00596-t001:** Primary outcomes. Ranges of rates and data extracted from the review with the largest sample are reported. Columns demonstrate analyzed outcomes, number of reviews that provided relevant data on the outcome, range of extractable raw rates, the extracted rate from the largest sample analyzed for the outcome and range of extracted pooled rates. Results are extracted from all eligible systematic reviews. Results are extracted only from moderate and high-quality reviews. Bold denotes differences between the two tables.

Primary Outcomes	Number of Reviews	Range	Largest N, n (%)	Range of *p* (%)
Cesarean delivery ^1^	34 [14,15,16,17,19,20,21,22,23,24,25,26,27,28,29,30,31,32,33,34,35,36,38,40,41,42,43,44,46,47,48,50,51,52]	52.3–95.8%	1060 (54.8%)	48.3–92.2%
Vaginal delivery	26 [14,16,20,22,23,26,28,23,29,30,31,32,33,34,35,36,38,39,40,41,42,43,44,47,50,51,52]	4.2–44.7%	856 (44.7%)	1.1–35%
Preterm deliveries ^1^	32 [14,15,16,17,20,21,22,23,24,25,26,27,28,29,30,32,34,35,36,38,39,40,41,42,43,44,45,46,48,50,51,52]	14.3–63.8%	386 (20.6%)	14.3–61.2%
Preterm labors ^2,3^	9 [19,23,28,29,32,39,41,44,47]	22.7–32.2%	22 (22.7%)	NA
PROMs ^4,5^	12 [22,25,26,28,29,30,38,39,41,44,47,52]	6.6–38.5%	16 (12.7%)	NA
pPROMs ^3^	6 [14,20,23,32,40,48]	6.4–16.1%	28 (6.4%)	5–18.8%
Miscarriages ^6,7^	10 [14,17,23,26,31,34,41,42,45,50]	<2.5%	3 (0.8%)	NA
**Primary Outcomes**	**Number of Reviews**	**Range**	**Largest N, n (%)**	**Range of p (%)**
Cesarean delivery	14 [24,25,26,27,31,32,35,36,40,43,46,47,48,52]	52.3–94%	1060 (54.8%)	48.3–92.2%
Vaginal delivery	11 [26,31,32,35,36,40,43,46,47,48,52]	6–44.7%	856 (44.7%)	1.1–35%
Preterm deliveries	13 [24,25,26,27,32,35,36,40,43,45,46,48,52]	20.6–63.8%	386 (20.6%)	16.9–61.2%
Preterm labors ^3^	2 [32,47]	NA	NA	NA
PROMs ^4^	5 [25,26,36,47,52]	NA	5 (5.3%)	NA
pPROMs ^3^	3 [32,40,48]	6.4–16.1%	28 (6.4%)	5–18.8%
Miscarriages ^7^	3 [26,31,45]	≤2%	1 (2%)	NA

Abbreviations: n, number of included participants with the outcome; N: sample screened for the primary outcome; *p*, pooled proportion; PROMs, premature rupture of membranes; pPROMs, preterm premature rupture of membranes; NA, Non-applicable. ^1^ One review presented the outcome of interest as range, which could not be applied to our calculated range [17]. ^2^ One review reported on spontaneous preterm labor (25(3.9%)) [23] and another one on preterm labor as indication for cesarean delivery (4 (11.8%)) [44], which could not be included in our calculated ranges. ^3^ One review reported a composite outcome of pPROM and preterm labor (24 (9%)) [32], which could not be included in our calculated ranges. ^4^ One review reported only study (not participant) rates [25], which could not be applied to our calculated ranges. ^5^ One review reported rates of PROM only among complicated pregnancies (7 (16%)) [30], while another one reported PROM only among cesarean delivery indications (9 (26.5%)) [44], which could not be included in our calculated ranges. ^6^ One review reported rates of miscarriages for first (5(16.1%)) and second (2 (3.6%)) trimester infections [23], which could not be included in our calculated ranges. ^7^ One review reported a composite outcome of miscarriage/termination (4 (1.4%)) [31] and another one on miscarriage/intrauterine fetal death (1 (3%)) [42], which could not be included in our calculated ranges.

**Table 2 ijerph-18-00596-t002:** Symptoms of infected mothers. Columns demonstrate analyzed outcomes, number of reviews that provided relevant data on the outcome, range of extractable raw rates, the extracted rate from the largest sample analyzed for the outcome and range of extracted pooled rates. Results are extracted from all eligible systematic reviews. Results are extracted only from moderate and high-quality reviews. Bold denotes differences between the two tables.

Symptoms	Number of Reviews	Range	Largest N, n (%)	Range of *p* (%)
Asymptomatic ^1^	13 [17,23,24,27,28,30,32,34,35,36,41,42,50]	7.5–32.6%	253 (21%)	9–30.9%
Fever ^1,2^	30 [14,16,17,19,20,21,22,23,24,25,26,27,28,29,30,31,32,34,35,36,40,41,43,44,45,46,47,48,50,52]	32.8–87.5%	2733 (32.8%)	39.7–86.5%
Cough	27 [14,19,20,21,22,23,24,25,26,27,28,29,30,31,32,34,35,36,40,41,43,44,45,47,48,50,52]	29–70%	3432 (41.3%)	31–71.4%
Fatigue ^3,4,5^	17 [14,21,22,24,25,26,27,28,29,30,31,32,35,41,44,47,50]	6.5–30.3%	192 (30.3%)	6–30.3%
Myalgia ^4,5,6^	21 [14,20,21,22,23,24,25,26,27,28,29,30,31,32,35,40,41,43,44,47,50]	6–24.4%	1411 (23.2%)	6–20.8%
Dyspnea ^4,7^	20 [14,23,24,25,26,28,29,31,32,34,35,40,41,43,44,45,47,48,50,52]	7.3–35.6%	1928 (23.6%)	8.9–18.8%
SOB ^7^	8 [14,20,21,22,27,28,30,31]	6.5–40.6%	789 (40.6%)	16.1–34.4%
Headache	5 [14,20,21,27,29]	3.3–27%	88 (13.1%)	9.3–15%
Sore throat ^4,8^	17 [14,20,22,23,24,25,26,28,30,31,32,35,41,43,44,50,52]	3.4–24%	78 (11.7%)	3–22.6%
Diarrhea ^9^	19 [14,20,23,24,26,27,28,29,30,31,32,35,40,41,43,44,47,50,52]	3.5–12%	659 (8.7%)	3–15.6%
Loss of smell/taste ^10^	2 [27,40]	NA	NA	NA
**Symptoms**	**Number of Reviews**	**Range**	**Largest N, n (%)**	**Range of *p* (%)**
Asymptomatic	5 [24,27,32,35,36]	8–32.6%	253 (21%)	9–30.9%
Fever	15 [24,25,26,27,31,32,35,36,40,43,45,47,48,52]	32.8–78%	2733 (32.8%)	39.7–86.5%
Cough	14 [24,25,26,27,31,32,35,36,40,43,45,47,48,52]	34–70%	3432 (41.3%)	36.8–71.4%
Fatigue ^3.4,5^	9 [24,25,26,27,31,32,35,40,43,47]	9.5–18.5%	101 (18.5%)	6–18.5%
Myalgia ^4,5,6^	10 [24,25,26,27,31,32,35,40,43,47]	6–24.4%	1411 (23.2%)	6–9.9%
Dyspnea ^4,7^	12 [24,25,26,31,32,35,40,43,45,47,48,52]	7.3–35.6%	1928 (23.6%)	8.9–18.8%
SOB ^7^	2 [27,31]	NA	789 (40.6%)	NA
Headache	1 [27]	NA	92 (14.4%)	NA
Sore throat ^4,8^	8 [24,25,26,31,32,35,43,52]	3.4–22.2%	10 (3.4%)	3–22.6%
Diarrhea ^9^	10 [24,26,27,31,32,35,40,43,47,52]	4–10.4%	659 (8.7%)	3–15.6%
Loss of smell/taste ^10^	2 [27,40]	NA	NA	NA

Abbreviations: n, number of included participants with the outcome; N, sample screened for the outcome; *p*, pooled proportion; SOB, shortness of breath; NA, Non-applicable. ^1^ One review presented the outcome of interest only as range [17], which could not be applied to our calculated ranges. ^2^ One review reported on a composite outcome of fever and respiratory symptoms [16], which could not be applied to our calculated ranges.^3^ Two reviews reported on a composite outcome of fatigue or malaise [22,35], which could not be applied to our calculated ranges. ^4^ One review reported only study (not participant) rates [25], which could not be applied to our calculated ranges. ^5^ One review reported on a composite outcome of myalgia, malaise or fatigue [32], which could not be applied to our calculated ranges. ^6^ One review reported on a composite outcome of myalgia, limb or joint pain [27], which could not be included in our calculated ranges. ^7^ Two reviews reported on a composite outcome of dyspnea and shortness of breath [28,31], which could not be applied to our calculated ranges. ^8^ One review reported on a composite outcome of sore throat and nasal congestion (mild respiratory symptoms) [32], which could not be applied to our calculated ranges. ^9^ Two reviews reported on gastrointestinal symptoms [29,47], and two others on composite outcomes of diarrhea and gastrointestinal symptoms [31], and diarrhea and abdominal pain [32], which could not be applied to our calculated ranges. ^10^ One review reported on loss of smell/taste rate of 30.6% [27], and another one only on ageusia (7.7%) [40].

**Table 3 ijerph-18-00596-t003:** Laboratory parameters of infected mothers. Ranges of rates and data extracted from the review with the largest sample are reported. Columns demonstrate analyzed outcomes, number of reviews that provided relevant data on the outcome, range of extractable raw rates, the extracted rate from the largest sample analyzed for the outcome and range of extracted pooled rates. Results are extracted from all eligible systematic reviews. Results are extracted only from moderate and high-quality reviews. Bold denotes differences between the two tables.

Laboratory Parameters	Number of Reviews	Range	Largest N, n (%)	Range of *p* (%)
Leukocytosis	7 [14,20,23,28,29,40,49]	13.9–45.8%	53 (13.9%)	27.4–33%
Leukopenia	3 [14,23,52]	<45.3%	53 (13.9%)	<45.3%
Lymphocytopenia	22 [14,19,20,21,23,24,25,27,28,29,31,32,34,35,36,40,43,47,48,49,50,52]	14–68.2%	262 (33.6%)	34.2–69.6%
cThrombocytopenia	6 [20,27,28,29,40,50]	1–44%	36 (8.4%)	3.2–8.2%
Elevated CRP ^1^	15 [14,23,24,27,28,29,31,32,34,40,43,47,49,50,52]	18.7–81.3%	174 (40.8%)	48–69%
Elevated PCT ^1^	2 [27,40]	NA	60 (23%)	NA
Elevated D-dimers	4 [23,27,49,50]	22.3–84.6%	86 (22.3%)	82–84.6%
Abnormal LFTs ^2^	7 [23,31,32,40,44,47,48]	8–27.3%	51 (10.4%)	10.6–29.6%
Elevated ALT	7 [14,20,29,44,47,50,52]	5.5–21%	21 (5.5%)	18.8–22.3%
Elevated AST	9 [14,20,21,27,29,44,47,50,52]	5.7–25%	22 (5.7%)	16–23.3%
**Laboratory Parameters**	**Number of Reviews**	**Range**	**Largest N, n (%)**	**Range of *p* (%)**
Leukocytosis	2 [40,49]	NA	50 (19.9%)	27.4–33%
Leukopenia	1 [52]	NA	0 (0%)	NA
Lymphocytopenia	13 [24,25,27,31,32,35,36,40,43,47,48,49,52]	29–68.2%	262 (33.6%)	35–69.6%
Thrombocytopenia	2 [27,40]	2.7–8.4%	36 (8.4%)	3.2–8.2%
Elevated CRP ^1^	9 [24,27,31,32,40,43,47,49,52]	40.8–70.3%	174 (40.8%)	48–69%
Elevated PCT ^1^	2 [27,40]	NA	60 (23%)	NA
Elevated D-dimers	2 [27,49]	NA	77 (84.6%)	82–84.6%
Abnormal LFTs ^2^	5 [31,32,40,47,48]	8–27.3%	51 (10.4%)	10.6–29.6%
Elevated ALT	2 [47,52]	NA	NA	NA
Elevated AST	3 [27,47,52]	NA	48 (15.1%)	16–23.3%

Abbreviations: n, number of included participants with the outcome; N: sample screened for the outcome; *p*, pooled proportion; CRP, C-reactive protein; PCT, Procalcitonin; LFTs, Liver Function Tests; ALT, Alanine aminotransferase; AST, Aspartate aminotransferase; NA, Non-applicable. ^1^ One review studied a composite outcome of elevated CRP or PCT (144(41%)), which could not be included in our calculated range [27]. ^2^ Encompasses the outcomes: “transaminitis” [23], “elevated AST or ALT” [31,32], “abnormal LFTs” [40], “elevated ALT and AST” [44,47], and “elevated liver enzymes” [48].

## Data Availability

The data presented in this study are available on request from the corresponding author.

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
