# Peer review of "Maternal and Neonatal Characteristics and Outcomes of COVID-19 in Pregnancy: An Overview of Systematic Reviews"

_ijerph, 2021, doi:10.3390/ijerph18020596_

Round 1

Reviewer 1 Report

The study is presented as a systematic review of Maternal and neonatal characteristics and outcomes of COVID-19 in pregnancy.

The aim of this study is to synthesize and evaluate the published systematic reviews on COVID-19 in pregnancy. The study has been carried out according to standards and has been described very well. As the topic is very interesting, the manuscript presents several limitations that in the current state prevent its publication.

Abstract

Point 1: The aim does not match what was stated in the introduction section.

Introduction

Point 2: The aim should be reformulated, since the effect of SARS-CoV-2 in infected pregnant women and their neonates, cannot be analyzed, but rather describes the obstetric-perinatal and neonatal care of infected pregnant women and their newborns during the pandemic. Only the effect could be analyzed by meta-analysis, which the authors have not performed.

Methods

Point 3: In PROSPERO, the review question was made by the PECOS format. In order to improve and clarify this section, I recommend specify it, and put in the same table the strategy a definition, and search terms in each database (Pubmed, Scopus and Cochrane Database). Please, feel free to use this example: doi:10.3390/ijerph17207405.

Point 4: In my opinion, the third inclusion criteria override the third exclusion criteria, so what is not included cannot be excluded.

Point 5: The authors should clarify why in the eligibility phase, they apply the exclusion criterion of No systematic reviews or protocols (types of studies), when they should have applied it in the screening phase. In addition, another duplicate article appears in this phase, when it should have been detected previously.

Results

Point 6: The format of the tables is not correct and are very difficult to read it. The title of the table must be put first, then the table, and finally the abbreviations. Put the table 1 in line 237 and correct the rest.

In order to clarify these tables, it could be possibly creating another table with the abbreviations. Or other possibilities. Please find an alternative if it is possible.

Point 7: I recommend that the quality of the studies be taken into account when including them in the review. According to the authors, only 3 of the reviews included high-quality articles (lines 175-177). Under deep justification, articles of moderate quality could be considered (13), reaching a total of 16 valid articles. Thus, this is one of the main concerns of this review, because the results of the articles with low quality interfere with the overall results. I recommend removing them, and only taking the comments above into account.

Discussion and Conclusion

Point 8: Taking into account the previous comments, the discussion and conclusions should be adapted to the new results.

Author Response

We are submitting for consideration our revised manuscript (ijerph-1037757) entitled: "Maternal and neonatal characteristics and outcomes of COVID-19 in pregnancy: an overview of systematic reviews”. We would like to thank the reviewers and the Editor of the Journal for taking the time and effort to assess our original submission so meticulously. We have taken into account all of their comments and recommendations and we have modified our paper accordingly. All manuscript changes have been highlighted in red color. Detailed replies to the reviewers’ comments are provided below:

Answers to reviewer no 1:

Point 1: “The aim does not match what was stated in the introduction section.”

Authors’ reply:We thank the reviewer for his accurate point. We have changed the aim accordingly; it now reads: The aim was to describe the obstetric-perinatal and neonatal outcome of infected pregnant women and their newborns during the pandemic”. The aim in the first paragraph of the “discussion” section has been modified accordingly (see: page 1, lines 19-21; page 2, lines 67-69; page 14, lines 479-481)

Point 2:The aim should be reformulated, since the effect of SARS-CoV-2 in infected pregnant women and their neonates, cannot be analyzed, but rather describes the obstetric-perinatal and neonatal care of infected pregnant women and their newborns during the pandemic. Only the effect could be analyzed by meta-analysis, which the authors have not performed.”

Authors’ reply:We thank the reviewer for his accurate point. We have changed the aim accordingly; it now reads: The aim was to depict the obstetric-perinatal and neonatal outcome and care of infected pregnant women and their newborns during the pandemic”. (see: page 1, lines 19-21; page 2, lines 67-69; page 14, lines 479-481)

Point 3:“In PROSPERO, the review question was made by the PECOS format. In order to improve and clarify this section, I recommend specify it, and put in the same table the strategy a definition, and search terms in each database (Pubmed, Scopus and Cochrane Database). Please, feel free to use this example: doi:10.3390/ijerph17207405.”

Authors’ reply: Thank you for your useful suggestion. We have formulated a table (Supplementary Table 1:  PECO strategy: category, definition, and search terms in databases) according to the example you kindly provided us (doi:10.3390/ijerph17207405), that clarifies the PECOS format and includes the strategy, definition and search terms in each database. The former Supplementary Table 1 has been renamed as Supplementary Table 2.

Point 4:“In my opinion, the third inclusion criteria override the third exclusion criteria, so what is not included cannot be excluded.”

Authors’ reply:We agree. We have removed the inclusion criterion no 3: “studies with human participants”. (page 2, line 74)

Point 5:“The authors should clarify why in the eligibility phase, they apply the exclusion criterion of No systematic reviews or protocols (types of studies), when they should have applied it in the screening phase. In addition, another duplicate article appears in this phase, when it should have been detected previously.”

Authors’ reply: Thank you for your thoughtful comment. The exclusion criterion of No systematic reviews or protocols (types of studies) was applied in the eligibility phase in order to prevent exclusion of systematic reviews, which were not easily identifiable by their title or abstract. The duplicate was also not identified by the system during the initial deduplication process and was therefore discovered in the eligibility phase. However, according to your suggestion with which we totally agree, we have modified our flow diagram (Figure 1), so that the 343 “not reviews/ not systematic reviews” as well as the 3 “protocols only” are excluded in the screening phase (“records excluded” (n = 444))and the one duplicate is excluded during the deduplication process (“records after duplicates removed” (n = 515). Lines 142-145 of the results section have been modified accordingly and now read: “After removal of duplicates, 515articles were screened based on title and abstract only, with 444of them being excluded. Full texts were screened from 71studies. Of these, 32were excluded”. (see page 4, lines 155-157, Figure 1)

Point 6:“The format of the tables is not correct and are very difficult to read it. The title of the table must be put first, then the table, and finally the abbreviations. Put the table 1 in line 237 and correct the rest. In order to clarify these tables, it could be possibly creating another table with the abbreviations. Or other possibilities. Please find an alternative if it is possible.”

Authors’ reply: Thank you for your correct observation. According to your suggestion, the title of each table has been put first, then the table, and finally its corresponding abbreviations and footnotes. These amendments have been applied in all tables 1a, 1b, 2a, 2b, 3a, 3b. In order to make our tables more easily readable, and in accordance with your recommendation, we have created a separate table (Supplementary Table 3), that summarizes all the abbreviations used in tables.

Point 7:“I recommend that the quality of the studies be taken into account when including them in the review. According to the authors, only 3 of the reviews included high-quality articles (lines 175-177). Under deep justification, articles of moderate quality could be considered (13), reaching a total of 16 valid articles. Thus, this is one of the main concerns of this review, because the results of the articles with low quality interfere with the overall results. I recommend removing them, and only taking the comments above into account.”

Authors’ reply: Thank you for a valuable recommendation that definitely improved our work. We have initially provided the results of all studies considered eligible according to our predefined protocol. However, in order to conform with your valuable suggestion, we have conducted a separate analysis and have presented the same results, taking into account only studies of “moderate” and “high quality” as these resulted from quality assessment using the AMSTAR-2 tool. In order to serve this purpose, we have added the following:

Abstract: The following sentence has been added in the “Methods” section: “A separate analysis, including only moderate and high-quality systematic reviews, was also conducted.” (page 1, lines 24-25)In the “Results” section, we have added the following sentence: “After accounting for quality of studies, ranges of our primary outcomes remained almost unchanged, while among our secondary outcomes, maternal ICU admission (3%-10%) and mechanical ventilation rates (1.4%-5.5%) were found relatively lower.” (page 1, lines 34-37)

Methods: The following sentences, along with the recommended justification for inclusion of “moderate quality” studies, have been added: “The same data were also presented after excluding “very low” or “low quality” studies and after taking into account only “moderate” and “high quality” studies, as these resulted from the quality assessment using the AMSTAR-2 tool. In this second analysis, studies of “moderate quality” were also included, taking into consideration that the majority of available systematic reviews were primarily based on preliminary data and were also designed and conducted at a time, at which urgent answers were needed.”(page 4, lines 148-153)

Results:  Sentences presenting the results of our separate analysis have been inserted throughout the whole “Results” section.  (page 6, lines 204-206, lines 215-216, lines 225-227, lines 232-233; page 7, lines 240-242, lines 246-247, lines 255-256; page 12, lines 365-366, line 369, lines 376-378, lines 384-386, lines 393-395, lines 400-401; page 13, lines 412-414, lines 418-420, lines 435-437; page 14, lines 443-445, lines 452-454, lines 461-463) For the reporting of our primary outcomes, the symptoms and laboratory findings of infected pregnant women, we have further added the tables 1b, 2b, and 3b, respectively. Our prior tables 1, 2, and 3 have been renamed as tables 1a, 2a, and 3a, respectively.

Point 8:“Taking into account the previous comments, the discussion and conclusions should be adapted to the new results.”

Authors’ reply: Thank you for meticulously reviewing our manuscript and for suggesting ways for its enhancement. Taking into account your previous comments and the results of this separate analysis, we have added the following sentences:

Discussion:The following text has been added in the first paragraph of the “discussion” section:“After taking into account only the rates provided by moderate and high-quality studies, CS rates were between 52.3% - 94%, preterm deliveries were between 20.6% - 63.8% and pPROM between 6.4% - 16.1%. Fever (32.8% - 78%) and cough (34% - 70%) were again the most frequently recorded symptoms, while maternal ICU admission (3% - 10%) and mechanical ventilation rates (1.4% - 5.5%) were found somewhat lower than the analysis of all studies. Maternal mortality did not change (<2%), while neonatal ICU admission and mortality rates were 10% - 76.9% and ≤ 2.4%, respectively. Neonatal PCR positivity rates ranged between 2% and 7%. Finally, with regard to the remainder of our secondary outcomes, mild differences were observed except “fatigue” (6.5% - 30.3% versus 9.5% - 18.5%), “lymphocytopenia” (14% - 68.2% versus 29% - 68.2%), “thrombocytopenia” (1% - 4.4% versus 2.7% - 8.4%) and “elevated CRP” (18.7% - 81.3% versus 40.8% - 70.3%).” (page 15, lines 489-499) Ranges were added in page 15, lines 483-485 to facilitate comparison.

The following sentence has been added in the “Limitations and strengths” paragraph: We also conducted a separate analysis, extracting results only from moderate and high-quality studies.” (page 17, lines 611-612)

Conclusions: The following sentence has been added: “After taking into account only moderate and high-quality studies, ranges of our primary outcomes remained unchanged, while among our secondary outcomes, maternal ICU admission and mechanical ventilation rates were found relatively lower.” (page 18, lines 625-628)

Reviewer 2 Report

This is a really important synthesis of all of the published systematic reviews about COVID-19 and pregnancy and will be a valuable source of information for many readers.

Please consider the following:

  1. In the abstract at lines 25 - 26 it needs to be made clear that these delivery rates are across all gestations so includes preterm and term.
  2. Similarly, in the abstract, at line 33 it needs to be made clear that the hight ICU admission rates are for symptomatic women with confirmed infection - this is mentioned in the methods but needs to be clarified here for all those readers who will glance at the abstract and perhaps not read the paper in detail.
  3. As with many papers there is some confusion between COVID-19 and SARS-CoV-2 infection - here it relates to asymptomatic women in some of the reviews considered who presumably have tested positive for virus but lack symptoms so don't technically have COVID-19. This is important to clarify in the Materials and Methods eligibility criteria section.
  4. Page 4, line 150 - the last search date is stated to be July 8 2020 but elsewhere (line 82 on page 3) this is stated to be Sept 10 2020.
  5. Page 4/line 65 and page 5/line 166 please clarify what the rates are - presumably of infection but this needs to be re-written to make clearer.
  6. Page 5/lines 181-182, as per the abstract it needs to be clarified that this is for preteen and term birth.
  7. In section 3.3.4 while the authors allude to the challenges around terminology across the systematic reviews included in their analysis they need to make sure the is sealer fo re reader. On page 7, line 235 'threatened abortions' presumably relates to threatened miscarriage but this needs to be clarified.
  8. Table 2 - SOB should be spelt out in a footnote even though the abbreviation is provided on page 3, line 108 it will be helpful to readers to include it here as well.
  9. Section 3.4.3 on page 10 - perhaps the heading should be 'Radiological imaging findings' for clarity. 
  10. 'overcome' at line 325 and 339 on page 11 and line 374 on page 12 should be replaced with exceed.
  11. Page 11, line 337 - insert 'case of' ahead of FGR.
  12. Page 12, line 360, change 'composing it' to 'composite'
  13. Page 12, line 368 - add 'these' before admissions for clarity.
  14. Page 12, line 378, insert 'that were' before PCR.
  15. Sections 3.4.15 and 3.4.16 - in the opening sentence of 3.4.15 the authors refer to 'and/or immunoglobulin positive' yet section 3.4.16 is about serum antibody positivity so this needs to be resolved and clarified
  16. Page 12, line 385, remove 'neonatal'
  17. Page 12, line 391, insert 'umbilical' before cord
  18. Page 12, line 393, add 'positive' before cord
  19. Page 13, line 394/395 - It is no clear what 'two of them reviewed......' is trying to say.
  20. Page 13, line 397 - replace 'totally' with 'a total of'
  21. Page 13, line 412 - replace 'in' with 'at'
  22. Page 13, lines 436/437 - sentence beginning 'Thus, safe....' needs re-writing for clarity
  23. Page 14, line 487 - realise 'embryo' with 'offspring' or similar
  24. Page 15, line 490 - replace 'till today' with 'up until the time of writing' - this will need to be checked and updated as needed.
  25. Page 15, line 525 - this needs clarification so as to not mislead the reader that mortality rates are high.
  26. Page 15, line 527 - replace 'rest of' with 'remainder of the'; replace 'with' what 'are of'
  27. Some of the points on page 14, lines 459 - 469 need to be made earlier - the lower incidence of symptomatic disease in pregnant women but the potential poor outcomes in pregnant women who do have symptoms especially should they require hospitalisation due to the severity of symptoms otherwise the overview is unbalanced.

Author Response

We are submitting for consideration our revised manuscript (ijerph-1037757) entitled: "Maternal and neonatal characteristics and outcomes of COVID-19 in pregnancy: an overview of systematic reviews”. We would like to thank the reviewers and the Editor of the Journal for taking the time and effort to assess our original submission so meticulously. We have taken into account all of their comments and recommendations and we have modified our paper accordingly. All manuscript changes have been highlighted in red color. Detailed replies to the reviewers’ comments are provided below:

for the reviewer no2

Reviewer #2:

Point 1:“In the abstract at lines 25 - 26 it needs to be made clear that these delivery rates are across all gestations so includes preterm and term.”

Authors’ reply:Thank you for your insightful comment. We have rewritten line 25 in the abstract clarifying that the delivery rates presented refer to all gestations. The specific modification is: “Reported rates, regardingboth preterm and term gestations,varied…” (page 1, line 27)

Point 2:“Similarly, in the abstract, at line 33 it needs to be made clear that the hight ICU admission rates are for symptomatic women with confirmed infection - this is mentioned in the methods but needs to be clarified here for all those readers who will glance at the abstract and perhaps not read the paper in detail.”

Authors’ reply:Thank you for your observation. The sentence at line 33 in the abstract has been modified in order to highlight that these ICU admission rates are for symptomatic women as follows: “In cases of symptomatic women with confirmed infection,high maternal and neonatal ICU admission rates should raise some concerns.” (page 1, lines 38-39)

Point 3:As with many papers there is some confusion between COVID-19 and SARS-CoV-2 infection - here it relates to asymptomatic women in some of the reviews considered who presumably have tested positive for virus but lack symptoms so don't technically have COVID-19. This is important to clarify in the Materials and Methods eligibility criteria section.”

Authors’ reply:Thank you for your remark. Since our overview includes reviews with asymptomatic SARS-CoV-2 infected women as participants, the 4th eligibility criterion in line 66 has been changed to “studies on characteristics and outcomes of pregnant or recently pregnant (post-partum, post-abortion, post-miscarriage) women with RT-PCR confirmed or suspected (based on clinical and imaging findings) SARS-CoV-2 infection, characteristics and outcomes of their neonates or the potential of SARS-CoV-2 vertical transmission;’’. The term “COVID-19” has been replaced with the more accurate “SARS-CoV-2 infection”. (page 2, line 76)

Point 4:“Page 4, line 150 - the last search date is stated to be July 8 2020 but elsewhere (line 82 on page 3) this is stated to be Sept 10 2020.”

Authors’ reply:Thank you for your observation. The last search date of our overview was indeed performed on September 10, 2020. We have extracted the last search date of each one of the eligible systematic reviews included in our overview. As a result, July 8, 2020 corresponded to the most recent search date among the included systematic reviews and not to the last search date of our overview. In order to clarify this part, we have modified this sentence that now reads: “Among the eligible systematic reviews’ last search dates, the most recent one took place on July 8, 2020.”(page 5, lines 165-166)

Point 5:“Page 4/line 65 and page 5/line 166 please clarify what the rates are - presumably of infection but this needs to be re-written to make clearer.”

Authors’ reply:Thank you for your constructive remark. These rates reflect the proportion of women infected with SARS-CoV-2 during a specific trimester. For instance, the rates 5% and 6% reflect the proportions of women infected with SARS-CoV-2 during the first trimester of their pregnancy among all participants with available data on trimester of infection. In order to clarify this part, we have amended the text so that now reads: “Only two reviews (12.5%) clearly stated the inclusion of first-trimester pregnancies with SARS-CoV-2 infection, with rates of women that were infected during their first trimester of pregnancy in these two reviews being 5% and 6% [23,34]. Seven reviews (46.7%) clearly stated the inclusion of second trimester pregnancies with SARS-CoV-2 infection, with rates of women that were infected during their second trimester varying between 1% and 10% [20,23,34-36,42,44].”(page 5, lines 180-185)

Point 6:“Page 5/lines 181-182, as per the abstract it needs to be clarified that this is for preteen and term birth.”

Authors’ reply:Thank you for your constructive remark. We have rewritten the first sentence of our ‘Primary outcomes’- ‘Modes of delivery’ section making it clear that we are referring to both preterm and term birth. The modification is: “Thirty-five reviews exhibited data on selected delivery modes for both preterm and term gestations.” (page 6, lines 198-199) Our following primary outcomes are specifically devoted to preterm delivery, preterm labor, PROM and pPROM, miscarriages and abortions.

Point 7:“In section 3.3.4 while the authors allude to the challenges around terminology across the systematic reviews included in their analysis they need to make sure the is sealer fo re reader. On page 7, line 235 'threatened abortions' presumably relates to threatened miscarriage but this needs to be clarified.”

Authors’ reply:Thank you for your thoughtful comments. We have inserted the following explanatory sentence: “(Nonetheless, the term “threatened abortions” presumably relates to “threatened miscarriages”.)” (page 7, line 265)

Point 8:Table 2 - SOB should be spelt out in a footnote even though the abbreviation is provided on page 3, line 108 it will be helpful to readers to include it here as well.”

Authors’ reply:Thank you for your observation. In accordance with your suggestion, we have spelt the abbreviation “SOB, shortness of breath” in a footnote under tables 2a and 2b. (page 10, line 321) In order to assist the readers in the reading of all tables, we have created an additional table (Supplementary table 3), that summarizes and explains all the abbreviations used in tables.

Point 9:“Section 3.4.3 on page 10 - perhaps the heading should be 'Radiological imaging findings' for clarity.”

Authors’ reply:Thank you for your comment. The heading has been replaced according to suggestion. (page 12, line 360)

Point 10:“'overcome' at line 325 and 339 on page 11 and line 374 on page 12 should be replaced with exceed.”

Authors’ reply:Thank you for your useful note. We have made the replacement in all the three places you suggested: “Taking into account 27 relevant reviews, maternal death rates did not exceed2% [14,17-23,25,27-32,34-36,40,42,43,45,47,48,50-52].” (page 12, line 388) / “Reported stillbirth rates of 26 reviews did not exceed 2.5% [16,17,19,20,23-29,32,34-36,40-45,47,48,50-52].” (page 13, line 407) / “Based on the remaining 27 studies, neonatal mortality did not exceed3% (1/29).” (page 14, line 450)

Point 11:“Page 11, line 337 - insert 'case of' ahead of FGR.”

Authors’ reply:Thank you for your suggestion. We have amended the sentence accordingly: “Data on FGR are limited with only 3 reviews focusing on this outcome [14,45,48]. Reported rates were 0% (pooled: 0% (0-22, I2= 0%), 1.2% (1/86, 0-6.3) and 9% (corresponding to just 1 case ofFGR).” (page 13, line 404)

Point 12:“Page 12, line 360, change 'composing it' to 'composite'”

Authors’ reply:Thank you for your observation. We rephrased the sentence as you kindly noted. It is now as follows: “Twelve reviews included data on neonatal asphyxia [14,17,19,24,26,28,31,32,41,48,51,52] with one compositewith stillbirth and neonatal death as a single outcome [19].”(page 13, line 432)

Point 13:“Page 12, line 368 - add 'these' before admissions for clarity.”

Authors’ reply:Thank you for your useful note. We have made the suggested modification and the sentence is converted to: “Three large series demonstrated rates of 64.9% (137/211) [21], 11.3% (54/479) and 52 (96.3%) of theseadmissions being preterm neonates and SARS-CoV-2 negative) [23] and 27.3% (368/1348; pooled: 24.6% (14.3-36.6, I2= 94.9%)) [40].” (page 14, line 442)

Point 14:“Page 12, line 378, insert 'that were' before PCR”

Authors’ reply:Thank you for your kind comment. We have amended the sentence accordingly: “Twenty-eight reviews found neonates that werePCR positive to SARS-CoV-2 [15-17,19,20,23-35,37-39,41,43-45,50-52].” (page 14, line 456)

Point 15:“Sections 3.4.15 and 3.4.16 - in the opening sentence of 3.4.15 the authors refer to 'and/or immunoglobulin positive' yet section 3.4.16 is about serum antibody positivity so this needs to be resolved and clarified”

Authors’ reply:Thank you for your insightful comment. We adjusted section 3.4.15. accordingly, and now it states “Twenty-eight reviews found neonates that were PCR positive to SARS-CoV-2[15-17,19,20,23-35,37-39,41,43-45,50-52].” The phrase “and/or immunoglobulin” has been removed, since serum antibody positivity is discussed in the next section. Furthermore, the references used have been checked in order to accord with the modified statement and no changes were needed. (page 14, line 456)

Point 16:“Page 12, line 385, remove 'neonatal'”

Authors’ reply:Thank you for your accurate recommendation. The word “neonatal” has been removed. (page 14, line 456)

Point 17:“Page 12, line 391, insert 'umbilical' before cord”

Authors’ reply:Thank you for your suggestion. The word “umbilical”was added before cord, as suggested. (page 14, line 471)

Point 18:“Page 12, line 393, add 'positive' before cord”

Authors’ reply:Thank you for your comment. We inserted the word “positive”in front of cord. (page 14, line 473)

Point 19:“Page 13, line 394/395 - It is no clear what 'two of them reviewed......' is trying to say.”

Authors’ reply:Thank you for another careful observation. The term “reviewed” was misused and has been replaced with the more accurate term “reported”on page 14, line 475. Additionally, the word “reported” (page 14, line 476) has been replaced with the word “found”for fluency.  

Point 20:“Page 13, line 397 - replace 'totally' with 'a total of'”

Authors’ reply:Thank you for your remark. The word “totally” has been replaced with the phrase “a total of”on page 14, line 477.

Point 21:“Page 13, line 412 - replace 'in' with 'at'”

Authors’ reply:Thank you for your comment. We have replaced “in” with “at”on page 15, line 507.

Point 22:“Page 13, lines 436/437 - sentence beginning 'Thus, safe....' needs re-writing for clarity”

Authors’ reply:Thank you for your recommendation. The sentence on page 16, lines 531-532 has been changed to “Thus, safe conclusions on the increase ofpreterm births cannot still be extracted, yet parameters related to medical managementmight be involved.”

Point 23:“Page 14, line 487 - realise 'embryo' with 'offspring' or similar”

Authors’ reply:Thank you for your suggestion. The word “embryo” on page 17, line 583 has been replaced with the word “offspring”.

Point 24:“Page 15, line 490 - replace 'till today' with 'up until the time of writing' - this will need to be checked and updated as needed.”

Authors’ reply:Thank you for your constructive remark.  We have replaced “till today” with the more accurate term “up until the time of writing”.(page 17, 586-587) The validity of this sentence has been checked through a search on PubMed, using keywords such as “COVID-19”, “SARS-CoV-2”, “neonatal death”, “newborn’s death”, “neonatal mortality”. 

Point 25:“Page 15, line 525 - this needs clarification so as to not mislead the reader that mortality rates are high.”

Authors’ reply:Thank you for your insightful comment. The sentence on page 18, lines 622-623 has been modified to “the mortality trend was not clearly elucidated”. The word “high” has been removed and the phrase “not totally clarified” has been replaced with “notclearly elucidated” in order to avoid misinformation.

Point 26:“Page 15, line 527 - replace 'rest of' with 'remainder of the'; replace 'with' what 'are of'”

Authors’ reply:Thank you for your comment. We have made the modification and the sentence is converted to: “The remainder ofthe fetal/neonatal outcomes presented are of low incidence and were possibly related to prematurity.”(page 18, line 624)

Point 27:“Some of the points on page 14, lines 459 - 469 need to be made earlier - the lower incidence of symptomatic disease in pregnant women but the potential poor outcomes in pregnant women who do have symptoms especially should they require hospitalisation due to the severity of symptoms otherwise the overview is unbalanced.”

Authors’ reply:Thank you for dedicating time to review our paper and for suggesting ways towards its improvement. We understand that a relative comment should be made earlier. In order for the review not to be unbalanced, we have provided a separate second paragraph in the “Discussion” section regarding symptomatic pregnant women. This new paragraph reads as follows: “In general, the symptomatic infection in pregnant women seems to be of lower incidence compared to the general population. Nonetheless, in cases of pregnant women with symptoms like fever and cough, negative outcomes may be expected especially when hospitalization due to the severity of symptoms is required.” (page 15, lines 500-503) As far as prior lines 459-469 (now lines 555-564, page 16) are concerned, we decided not to rearrange them in an attempt to conform with protocol and aim of the study, in the context to ensure synchronization between the analysis of our primary and secondary outcomes in the “discussion” section and their presentation in the “results” section.

In conclusion, we hope that with these revisions, our work is felt appropriate to publish in the “International Journal of Environmental Research and Public Health”. All in all, we look forward to hearing from you and we would be pleased to answer any further questions and/or comments you may have.

Sincerely,

Charalampos Siristatidis

Associate Professor of Obstetrics and Gynecology / Assisted Reproduction

Assisted Reproduction Unit

Second Department of Obstetrics & Gynecology

"Aretaieion" Hospital

National and Kapodistrian University of Athens

Athens, Greece

Round 2

Reviewer 1 Report

The authors have made substantial changes to the manuscript based on the given comments and have addressed most of my concerns.  However, there are some aspects that we should improve.

Results and Discussion

Point 1: An explanation of the possible causes of the very wide confidence intervals and considerable heterogeneity (I2>70%) of the meta-analyses should be offered. The possible biases of the studies analysed should be studied in depth.

  1. Mode of delivery: c-section (27-43, I2= 91.4%); (8.9-36.6, I2= 89.4%)
  2. Preterm delivery: (13.2-20.9, I2= 71.5%); (3.3-7.5, I2= 82.3%); (8.3-35.8, I2= 87.4%)
  3. ICU admission: (1.8-7.1, I2= 93.6%)
  4. Mechanical ventilation: (0.8-5.2, I2= 93.5%); (1.5-7.7, I2= 90.2%)
  5. Maternal death: (0-0.7, I2= 80.2%)
  6. Apgar scores: (8.382-9.240, I2= 88.9%); (9.136-9.895, I2= 82.9%)
  7. NICU admissions: (14.3-36.6, I2= 94.9%)
  8. Neonatal PCR positivity: (0.4-4.7, I2= 59.8%): Not considerable but very important.

Point 2: In the discussion section the results should not be repeated, unless it is very precise. Try reducing the number of them between lines 468-488.

Point 3: In general, a final reflection should be made on how the health care provided has been able to affect compliance with the quality indicators of obstetric and perinatal care, causing situations that could have been classified within the framework of obstetric violence, hiding under the structural and assistance auspices of the SARS-CoV-2 pandemic. Feel free to read “Coxon K, Turienzo CF, Kweekel L, Goodarzi B, Brigante L, Simon A, et al. The impact of the coronavirus (COVID-19) pandemic on maternity care in Europe. Midwifery 2020:102779. doi:10.1016/j.midw.2020.102779.”

Author Response

We would like to thank the reviewer for taking the time and effort to assess our original submission so meticulously. We have taken into account all comments and recommendations and we have modified our paper accordingly. All manuscript changes have been highlighted in red color. Detailed replies to the reviewer’s comments are provided below:

Point 1: An explanation of the possible causes of the very wide confidence intervals and considerable heterogeneity (I2>70%) of the meta-analyses should be offered. The possible biases of the studies analysed should be studied in depth.

  1. Mode of delivery: c-section (27-43, I2= 91.4%); (8.9-36.6, I2= 89.4%)
  2. Preterm delivery: (13.2-20.9, I2= 71.5%); (3.3-7.5, I2= 82.3%); (8.3-35.8, I2= 87.4%)
  3. ICU admission: (1.8-7.1, I2= 93.6%)
  4. Mechanical ventilation: (0.8-5.2, I2= 93.5%); (1.5-7.7, I2= 90.2%)
  5. Maternal death: (0-0.7, I2= 80.2%)
  6. Apgar scores: (8.382-9.240, I2= 88.9%); (9.136-9.895, I2= 82.9%)
  7. NICU admissions: (14.3-36.6, I2= 94.9%)
  8. Neonatal PCR positivity: (0.4-4.7, I2= 59.8%): Not considerable but very important.

Answer:

We thank the reviewer for the comment. We have added the following sentence in the discussion section (limitations section):” we noted very wide confidence intervals and considerable heterogeneity (I2>70%) of the included studies, which are both linked with a high uncertainty of the evidence, through true effect / result. The reasons include various (usually small) sample sizes and characteristics of the populations studied, the number of studies combined and low precision of the individual study estimates. Thus, the results of the current study should be interpreted with caution, something that is sensible, due to the ongoing nature of the disease itself and the premature conduction of studies.”.

Point 2: In the discussion section the results should not be repeated, unless it is very precise. Try reducing the number of them between lines 468-488.

Answer:

We thank the reviewer for the comment. It now reads:

In this overview of systematic reviews, we tried to describe the obstetric-perinatal and neonatal outcome of infected pregnant women and their newborns during the SARS-COV-2 pandemic. After application of our inclusion criteria, we analyzed 39 reviews. CS rates were between 52.3%-95.8%, preterm deliveries were between 14.3%-63.8%, PROM between 5.3%-12.7% and pPROM between 6.4% - 16.1%. Fever and cough were the most frequently recorded symptoms, while maternal ICU admission  and mechanical ventilation rates were high. Neonatal ICU admission and mortality rates were 3.1%-76.9% and <3%, respectively. Neonatal PCR positivity rates ranged between 1.6% and 10%. The methodological quality of the reviews was assessed through the AMSTAR 2 checklist; interestingly, almost 2/3 of the reviews were assessed as being at “very low” or “low” quality, while only 7% of them were judged as being at “high” quality. After taking into account only the rates provided by moderate and high-quality studies, CS rates, preterm deliveries and pPROM were similar; fever and cough were again the most frequently recorded symptoms, while maternal ICU admission and mechanical ventilation rates were found somewhat lower than the primary analysis. Similarly, all the rest of the secondary outcomes remained unchanged.

Point 3: In general, a final reflection should be made on how the health care provided has been able to affect compliance with the quality indicators of obstetric and perinatal care, causing situations that could have been classified within the framework of obstetric violence, hiding under the structural and assistance auspices of the SARS-CoV-2 pandemic. Feel free to read “Coxon K, Turienzo CF, Kweekel L, Goodarzi B, Brigante L, Simon A, et al. The impact of the coronavirus (COVID-19) pandemic on maternity care in Europe. Midwifery 2020:102779. doi:10.1016/j.midw.2020.102779.”

Answer:

We thank the reviewer for the comment. We have added a couple of sentences in the discussion section, also adding a new reference (the proposed). It now reads:

«So far, data are not robust to lead to definite points and regulations. The SARS-CoV-2 pandemicimpact on maternity care in all over the world has yielded various responses, includinglabor care and choice of place of birth, increased risk adverse decisions, reduction in antenatal and postnatal ‘face to face’ care provision and various necessary adjustments with unknown impact on women and offspring’ wellbeing, or on women's experiences of birth [68].»